# ROR1 Expression and Its Functional Significance in Hepatocellular Carcinoma Cells

**DOI:** 10.3390/cells8030210

**Published:** 2019-03-02

**Authors:** Metin Cetin, Gorkem Odabas, Leon R. Douglas, Patrick J. Duriez, Pelin Balcik-Ercin, Irem Yalim-Camci, Abdulkadir Emre Sayan, Tamer Yagci

**Affiliations:** 1Department of Molecular Biology and Genetics, Gebze Technical University, Kocaeli 41400, Turkey; metincetin@gtu.edu.tr (M.C.); godabas@gtu.edu.tr (G.O.); pbalcik@gtu.edu.tr (P.B.-E.); iyalim@gtu.edu.tr (I.Y.-C.); 2Protein Core Facility, Cancer Research UK and Experimental Cancer Medicine Centres, University of Southampton, Southampton General Hospital, Southampton SO16 6YD, UK; L.R.Douglas@soton.ac.uk (L.R.D.); P.J.Duriez@soton.ac.uk (P.J.D.); 3Cancer Sciences Unit and Cancer Research UK Centre, University of Southampton, Southampton General Hospital, Somers Cancer Research Building, Mailpoint 824, Southampton SO16 6YD, UK

**Keywords:** HCC, EMT, ROR1, monoclonal antibody, hallmarks of cancer, drug resistance, drug efflux

## Abstract

Background: Hepatocellular carcinoma (HCC) is a common and deadly cancer; however, very little improvement has been made towards its diagnosis and prognosis. The expression and functional contribution of the receptor tyrosine kinase ROR1 have not been investigated in HCC before. Hence, we investigated the expression of ROR1 in HCC cells and assessed its involvement in hepatocarcinogenesis. Methods: Recombinant bacterial ROR1 protein was used as an immunogen to generate ROR1 monoclonal antibodies. *ROR1* transcript levels were detected by RT-qPCR and the protein expression of ROR1 in HCC was assessed by Western blotting by using homemade anti-ROR1 monoclonal antibodies. Apoptosis, cell cycle, trans-well migration, and drug efflux assays were performed in shRNA-ROR1 HCC cell clones to uncover the functional contribution of ROR1 to hepatocarcinogenesis. Results: New ROR1 antibodies specifically detected endogenous ROR1 protein in human and mouse HCC cell lines. ROR1-knockdown resulted in decreased proliferation and migration but enhanced resistance to apoptosis and anoikis. The observed chemotherapy-resistant phenotype of ROR1-knockdown cells was due to enhanced drug efflux and increased expression of multi-drug resistance genes. Conclusions: ROR1 is expressed in HCC and contributes to disease development by interfering with multiple pathways. Acquired ROR1 expression may have diagnostic and prognostic value in HCC.

## 1. Introduction

Liver cancer is the seventh most common cancer and fourth cause of cancer associated mortality world-wide [1]. Hepatocellular carcinoma (HCC) is the most common form of liver cancer. There are several risk factors for HCC such as chronic hepatotrophic virus (Hepatitis B virus and/or Hepatitis C virus) infections, excess alcohol consumption, aflatoxins, hemochromatosis, and obesity [2,3,4]. Despite advances in diagnostic and therapeutic options, incidence of HCC is still increasing in many countries, especially in the Western hemisphere [5]. Alterations of various pro-survival pathways, such as activated receptor tyrosine kinase (RTKs) signaling, play a fundamental role in the development of HCC [6]. Targeting cancer using anti-RTK antibodies became a therapeutic avenue for malignant tumors with aberrant RTK expression. Critically, the RTK of interest should be mutated or overexpressed in the context of intended cancer with a clear contribution to the specific attributes of carcinogenesis. Anti-ERBB2 blocking antibody Trastuzumab as used to treat breast and gastroesophageal cancers, and anti-VEGF (Bevacizumab) and anti-EGF (Cetuximab, Panitumumab) antibodies as used in colorectal cancer treatment are good examples that showed clinical benefit [7,8,9,10,11]. 

Receptor tyrosine kinase-like orphan receptor (ROR) is a family of RTKs consisting of ROR1 and ROR2. They contain four evolutionally conserved domains, namely the extracellular immunoglobulin-like (Ig-like), Frizzled, and Kringle domains, and the intracellular kinase domain. The name ROR originated from the absence of a known ligand and the presence of the kinase-like domain. Currently, Wnt5a is the proposed ligand of ROR1 and ROR2. cDNAs of ROR proteins were cloned from a human neuroblastoma cell line [12] and their roles in neurons, such as neurite growth and synapse formation, were well-established [13]. Among the ROR family, the association of ROR1 with cancer was first shown in B-cell malignancies [14,15,16]. In addition, aberrant expression of ROR1 and the consequences of this occurrence have been reported in pancreatic, breast, and ovarian cancers as well as melanoma [17,18,19]. Focused studies in breast cancer pointed to the role of ROR1 in enhanced tumor cell growth, epithelial-to-mesenchymal transition (EMT), and metastasis. ROR1-mediated PI3K/AKT/CREB signaling contributed to the growth and survival of breast cancer cells and this signaling activity was augmented upon stimulation by Wnt5a [18]. In addition, expression of ROR1 in breast cancer cell lines was associated with the EMT gene expression signature, and silencing of ROR1 inhibited metastasis of tumor xenografts in nude mice [20]. The role of ROR1 as a pseudokinase in proliferation and tumorigenesis was shown to be sustained by its trans-phosphorylation by cMET in gastric and non-small cell lung cancer cells [21]. However, kinase activity was attributed to ROR1 upon observation that the survival of lung adenocarcinoma cells was maintained through phosphorylation of c-SRC by ROR1 and activation of AKT in these cells [22]. 

ROR1 is expressed during embryogenesis, and its expression is only limited to very few adult tissues [23] not including liver [24]. ROR1 expression and its potential function in HCC have not been studied so far. Considering the previously proposed contribution of ROR1 to metastasis, the assessment of ROR1 involvement in EMT and tumorigenesis of HCC is of utmost importance. 

In this report, we investigated the expression and functional contribution of ROR1 to several key aspects of HCC biology. To achieve this aim, we generated two new monoclonal antibodies for ROR1. We found that ROR1 is expressed in human and mouse HCC cells, and that its expression is not restricted to metastatic/mesenchymal HCC cell lines. However, TGFβ-induced EMT reduced ROR1 protein abundance, and shRNA-mediated ROR1 knockdown altered EMT status of HCC cells along with decreased proliferation and migration, and increased resistance to apoptosis and anoikis. Critically, ROR1 knockdown epithelial HCC cells had altered uptake of chemotherapeutic agents, which made them resistant to chemotherapy-induced apoptosis. Our results suggest that ROR1 is a potential biomarker in HCC that may prognosticate therapy response.

## 2. Materials and Methods

### 2.1. Tissue Culture

PLC/PRF/5, HepG2, HuH7, SNU387, SNU423, SKHep1, Hepa1-6, S6-1B, S6-1D, Hek293T, and SP2/0 cell lines were purchased from the American Type Culture Collection (ATCC), routinely tested for mycoplasma contamination (MycoAlert, Lonza), and annually subjected to authenticity validation by STR analysis. Cells were grown in DMEM (RPMI for SNU387 and SNU423) supplemented with 10% FBS, Penicillin/Streptomycin (50 U/mL), 2 mM L-Glutamine, and 1% non-essential amino acids in a humidified CO_2_ (5%) incubator. TGF-β-induced EMT was achieved by seeding PLC/PRF/5 cells into 60 mm Petri dishes and treating them with 4 ng/mL TGF-β for 36 or 96 h. 

### 2.2. Generation of Monoclonal Antibody

Recombinant ROR1 expression and purification was performed as described in [25]. Briefly, considering the antigenicity index and potential cross reactivity with other proteins, a 500 amino acid (aa) region of ROR1, starting from codon 120, was cloned into pOPINF (#26042, Addgene) plasmid with a N-terminal 6XHis tag. This region contained the majority of the extracellular domain, the membrane spanning region, and a part of the kinase domain. Primers used were hROR1_120_pOPIN-F: 5′-AAGTTCTGTTTCAGGGCCCGAACCTCGACACCACAGACAC-3′ and hROR1-620_pOPIN-R 5′-ATGGTCTAGAAAGCTTTAATTGCGAGCTGCAAGGTCCTT-3′. Recombinant protein was expressed in Rosetta™ (DE3) pLacI cells (Novagen) and purified using Ni-NTA resin (Qiagen). Site-directed mutagenesis (QuikChange II Site-Directed Mutagenesis Kit, Agilent) was used to introduce a stop codon at the end of the extracellular region to create aa 120–400 of ROR1. The production of monoclonal antibodies was described before [26]. Briefly, 8–10-week-old BALB/c mice were immunized with 50 μg recombinant protein emulsified in Complete Freund’s Adjuvant (Sigma-Aldrich) and then the following injections were carried out every 3 weeks with recombinant protein mixed with Incomplete Freund’s Adjuvant (Sigma-Aldrich). Sera of the immunized and control mice were tested for reactivity against recombinant protein with indirect ELISA after immunizations. The animal with the highest immunoreactivity against recombinant ROR1 was further boosted three days before the fusion. The fusion of freshly isolated splenocytes with SP2/0 myeloma cells was performed as described previously [27]. After the fusion procedure, the cells were seeded in 96-well plates and then were selected first with HAT and then with HT media. After single cell subcloning of the hybridoma cells, specific clones were expanded in culture and hybridomas were stored in liquid nitrogen. Antibody isotype was determined by using the Pierce rapid antibody isotyping kit (Thermo Fisher Scientific) according to the manufacturer’s instructions. 

### 2.3. Transfections and Virus Work

Validated lentiviral ROR1 shRNA (TRCN0000002026, Sigma), lentiviral packaging mix (SHP001, Sigma), and pLKO.1 (Addgene #8453) were used to obtain lentiviral particles. Briefly, 1 μg of ROR1 shRNA or pLKO.1 plasmids and 5 μL of lentiviral packaging mix were mixed with PEI (Polysciences) transfection reagent in 500 μL OptiMEM (Thermo Fisher Scientific). The mixture of plasmids and transfection reagent was incubated for 5 min and then transfected into HEK293T cells. After 48 h, the supernatant of the HEK293T cells containing viral particles was collected, filtered, aliquoted, and stored at −80 °C. 

### 2.4. Generation of ROR1 Knockdown Stable Cells

PLC/PRF/5 and SNU387 cells were seeded into 6-well plates at a density of 40% of confluency. Next day, cells were transduced with 1 mL of viral particles in the presence of 8 μg/mL polybrene (Thermo Fisher Scientific). After 24 h, cells were split into 60 mm Petri dishes and selection of clones were started with 2 μg/mL puromycin (InvivoGen). Stable heterogeneous ROR1-knockdown cells were generated after 2 weeks. Decrease in ROR1 protein abundance was assessed using Western blotting. 

### 2.5. In Silico Analyses 

In order to determine the expression of ROR1 in HCC, a search at the “European Bioinformatics Institute (EMBL-EBI) Expression Atlas” website (https://www.ebi.ac.uk/gxa/home) for *ROR1* genes on the *Homo sapiens* dataset with “disease” and “pan-cancer analysis of whole genomes-liver” filters was used. The output consisted of multiple liver derived cancers such as cholangiocarcinoma and HCC and their normal counterpart tissues. The output consisting of 99 HCCs and 52 normal liver samples were downloaded and analyzed for statistical significance (using the student t-test) and plots drawn using Microsoft Excel (Office 10). 

### 2.6. Western Blotting and RT-qPCR

Western blotting was performed as described previously [26]. The primary antibodies used in this study and their dilutions were as follows: ROR1 (1/500, homemade IC5 or 5B3 clones), β-actin (1/5000), E-cadherin (1/1000, BD Transduction Laboratories), Vimentin (1/1000), PARP (1/1000, Cell Signaling), CK19 (1/1000, Santa Cruz Biotechnology), and His-tag (1/3000, Qiagen). After treatment of PVDF membranes (Thermo Fisher Scientific) with primary antibodies, HRP-conjugated secondary antibody (1/3000, Cell Signaling) and Amersham ECL Select (GE Healthcare) chemiluminescence substrate were used to visualize protein bands by using the ChemiDoc XRS system (Bio-Rad). RNA isolation, cDNA synthesis, and RT-qPCR were performed as described before [26]. Relative expression of *ROR1* mRNA in HCC cell lines was measured by normalizing *ROR1* expression to that of *GAPDH* and calculated with the 2^− ΔCt^ formula [ΔCt =Ct (ROR1) − Ct (GAPDH)]. Primers for RT-qPCR were designed using Primer-BLAST. Sequence of primers were as follows: *ROR1-F* 5′-GTTTCCCAGAGCTGAATGGA-3′ and *ROR1-R* 5′-GGATGTCACACAGATCAGACTT-3′; *GAPDH-F* 5′-GGCTGAGAACGGGAAGCTTGTCAT-3′ and *GAPDH-R* 5′-CAGCCTTCTCCATGGTGGTGAAGA-3′.

### 2.7. Immunoprecipitation 

An equal amount of total protein lysate from SNU387 cells was incubated overnight at 4 °C with both 5B3 and 1C5 anti-ROR1 monoclonal antibodies followed an incubation of the antigen-antibody complexes with anti-IgG antibody-coated magnetic beads (Invitrogen) for 1 h at room temperature. The eluted immune complexes were analyzed for reciprocal incubation of the other ROR1 antibody (e.g., pull down by 5B3, Western blot with 1C5 and vice versa) by Western blot.

### 2.8. Flow Cytometry

PLC/PRF/5 cells were incubated with 4 mM EDTA solution for 10 min to detach from tissue culture flasks. Cells were then washed with PBS and centrifuged at 300 G for 5 min. Then, cells were re-suspended at 1 × 10^6^/100 µL density in PBS and stained with 10 µg of 5B3 antibody for 1 h at 4 °C. After the incubation, cells were washed with PBS and centrifuged at 300 G for 5 min. Cells were then incubated with Alexa488 fluorescence antibody (1/400, Cell Signaling) for 1 h at 4 °C. After the secondary antibody, cells were washed with PBS and centrifuged at 300 G for 5 min and analyzed with Accuri C6 flow cytometry (BD) in the FL1 channel.

### 2.9. Functional Assays: Proliferation, Cell cycle, Apoptosis, Doxorubicin uptake, Migration, and Drug Resistance

Effects of ROR1 knockdown on proliferation of PLC/PRF/5 and SNU387 was detected by xCELLigence RTCA DP (ACEA Biosciences) with real-time analysis. PLC-pLKO, PLC-shROR1, SNU387-pLKO, and SNU387-shROR1 cells were seeded at a density of 5 × 10^3^ into E-Plate 16. Impedance based cell index value of the wells, indicating cell number, were recorded up to 48 h. ROR1-dependent proliferation of cells was compared with the normalized cell index values. 

To perform cell cycle analysis, 2 × 10^5^ PLC-pLKO, PLC-shROR1, SNU387-pLKO, and SNU387-shROR1 cells were trypsinized and fixed overnight in 70% ethanol at 4 °C. Next day, cells were treated with 100 µL RNase A (0.260 Knudson U) and 400 µL PI (50 μg/mL) for 1 h at 37 °C, and excess dye was washed and removed by centrifugation. Cells were then re-suspended in 400 µL cold PBS and analyses were performed using FACS Calibur (BD) in the FL3 channel. 

For anoikis analysis, 1.5 × 10^5^ PLC-pLKO, PLC-shROR1, SNU387-pLKO, or SNU387-shROR1 cells were seeded into 6-well ultra-low attachment plates (Corning). After 24 h, cells were centrifuged and incubated with 4 mM EDTA solution for 10 min in a tissue culture incubator to disassociate clusters. Cells were then washed with PBS and centrifuged at 300 G for 5 min. Finally, cells were re-suspended at a density of 1 × 10^6^ cells/mL in 1× binding buffer (10 mM HEPES/NaOH, pH 7.4; 140 mM NaCl; 2.5 mM CaCl_2_) and stained with 5 µL annexin V-APC and 5 µL PI (BD) for 15 min at room temperature. Anoikis was analyzed using Accuri C6 (BD) flow cytometry.

To observe pro-apoptotic effects of Oxaliplatin (Hospira, UK) and Doxorubicin (Sigma) on ROR1 knockdown cells, 1.5 × 10^5^ PLC-pLKO, PLC-shROR1, SNU387-pLKO, and SNU387-shROR1 cells were seeded into 6-well plates. After 24 h, cells were treated with 0, 200, and 400 µM of oxaliplatin for 48 h and with 0, 0.5, and 3 µg/mL of Doxorubicin for 24 h [28,29]. Cells were then collected and Western blotting analysis was performed to detect PARP cleavage as an indicator of caspase activity. Percent survival was calculated by quantifying uncleaved PARP (p116-PARP intensity) as compared to actin using ImageJ software. 

The drug efflux assay was performed by detecting retained (intracellular) Doxorubicin, which has fluorescence properties. For this purpose, 1.5 × 10^5^ PLC-pLKO, PLC-shROR1, SNU387-pLKO, and SNU387-shROR1 cells were seeded into 6-well plates. After 24 h, cells were treated with 0.5 µg/mL of Doxorubicin for 1 h, washed with PBS and further incubated with fresh media for another 4 h. After the incubation, cells were trypsinized and analyzed using FACS Calibur (BD) in the FL3 channel. Mean fluorescence intensity (MFI) was used to quantify retained Doxorubicin.

Migration capacities of the control and ROR1 knockdown cells were analyzed as described previously [30]. Briefly, 2.5 × 10^5^ PLC-pLKO, PLC-shROR1, SNU387-pLKO, and SNU387-shROR1 cells were seeded in 8 µm pore-size BD-Falcon 24 well Transwell inserts (BD). After 2 h, the media at the top chamber was aspirated and serum-free media added to the chamber to create a serum gradient. Cells were incubated for 24 h. Next, both sides of the insert chambers were fixed with acetone:methanol (50%:50%). The top of the chamber was stained with the eosin solution and the bottom of the chamber was stained with the DAPI solution. The cells at the top of the chamber were wiped with a cotton bud. Then, the photo of the each insert-well was taken by an Olympus CKX41 fluorescence microscope in UV channel and nuclei of the cells were counted using ImageJ software and macro as defined in [30].

The expression of genes associated with drug resistance in PLC-pLKO, PLC-shROR1, SNU387-pLKO, and SNU387-shROR1 cells were analyzed by RT-qPCR by using the primers listed in Table 1.

## 3. Results

### 3.1. ROR1 Is Expressed in HCC Cell Lines Irrespective of Their EMT Phenotype

ROR1 expression has been associated with mesenchymal/metastatic features of several cancers [20,31,32], but it has not been studied in HCC before. However, analysis of the RNA-seq data from the “EMBL-EBI Expression Atlas” database pointed out the upregulated expression of *ROR1* in HCC tumors compared to normal liver tissues (Figure 1a). Therefore, we performed an expression analysis in HCC cell lines to gain insight into whether ROR1 is aberrantly expressed in this cancer. Importantly, to better understand if ROR1 has any association with metastatic behavior of cancer cells, we used a panel of six HCC cell lines with well-established EMT status [33]. Our results revealed a differential expression of ROR1 of up to five-fold difference between cell lines (Figure 1b). Normal liver tissue was reported to be devoid of ROR1 expression, therefore the increased abundance of ROR1 may have a functional consequence towards HCC biology [24]. However, when EMT status of HCC cell lines is considered, we observed no apparent association of ROR1 with epithelial or mesenchymal features (Figure 1b). These results suggest that ROR1 does not associate with fully epithelial or fully mesenchymal states of the EMT program. Therefore, it worth it to study the involvement of ROR in other characteristics of EMT and tumorigenesis, such as resistance to apoptosis, anoikis, decreased proliferation, and enhanced migration.

### 3.2. Generation and Characterization of Anti-ROR1 Monoclonal Antibodies

RTK antibodies, especially the ones recognizing an extracellular epitope, may have therapeutic function [34]. Given the limitation of commercially available antibodies in terms of specificity and the potential of newly generated ones in targeting distinct isotypes of proteins, we performed hydrophobicity analysis and identified a segment of ROR1 protein (aa 120–620) that may serve as a good antigen (Figure 2a). This segment has been cloned, bacterially expressed, purified, and used as an antigen for immunization. Among ELISA reactive clones, two strong hybridomas (5B3 and 1C5) were chosen for further analysis. The isotype of both antibodies was determined as IgG1. Both antibodies recognized the full length recombinant protein (aa 120–620) and a truncated version (aa 120–400) as strongly as the His-tag antibody (Figure 2b). Recognition of the smaller antigen by these monoclonal antibodies indicates that their cognate epitopes are in the extracellular region (Figure 2a,b). Moreover, both antibodies were able to pull down endogenous ROR1 protein as shown using immunoprecipitation followed by Western blotting (Appendix A
Figure A1A). Additionally, new antibodies allowed detection of ROR1 protein using flow cytometry (Appendix A
Figure A1B) suggesting that they work under native (flow cytometry, immune-precipitation) and denaturing (Western blotting) conditions.

### 3.3. Monoclonal Antibodies Specifically Recognize Endogenous ROR1 in Human and Mouse HCC Cells

To see if the new anti-ROR1 antibodies detect endogenous ROR1 protein, we used a panel of human (Figure 3a) and mouse (Figure 3b) HCC cell lines. Our results suggest that all tested cells have ROR1 protein, albeit at varying amounts, and the new ROR1 antibodies can detect human and mouse ROR1 protein effectively. The discrepancy between RT-qPCR and Western blot data might be due to the fact that ROR1 primers were designed to detect all transcript variants; however, cell lines displayed differential expression of 130 kDa and 100 kDa ROR1 protein isoforms. Both N-glycosylated membrane isoform (130 kDa) and the 100 kDa isoform of ROR1 were recognized by both antibodies [15,35]. To validate the specificity of 5B3 and 1C5, we knocked down the membrane isoform of ROR1 in two human HCC cell lines (PLC/PRF/5 and SNU387) with high ROR1 protein expression. A specific band at ~130 kDa decreased significantly as detected with both new ROR1 antibodies upon *ROR1* downregulation (Figure 3C). These results confirm that HCC cells have varying amounts of ROR1 protein, the new antibodies are specific and they can recognize both human and mouse endogenous ROR1 effectively.

### 3.4. ROR1 Regulates EMT Phenotype in HCC Cells

The expression of ROR1 has been associated with EMT and cancer stem cell phenotypes in different cancers previously, but there is no data in the context of HCC [20,31,32]. However, our results suggest that such a correlation does not exist as two out of three mesenchymal HCC (SNU423 and SK-Hep1) cells had low ROR1 protein (Figure 3a). To investigate the regulation of ROR1 expression during EMT, we treated the epithelial HCC cell line, PLC/PRF/5, with TGF-β for 36 and 96 h to induce EMT. Canonical hallmarks of EMT such as vimentin upregulation and E-cadherin downregulation were observed in TGF-β treated PLC cells along with the decreased expression of ROR1 (Figure 4a). These results suggest that ROR1 expression associates with an epithelial rather than a mesenchymal phenotype in HCC. This led us to investigate the involvement of ROR1 in EMT. Comparison of ROR1-knockdown and control clones of PLC cells demonstrated a shift to the mesenchymal phenotype as ROR1 knockdown cells expressed less E-cadherin and cytokeratin 19 and increased vimentin (Figure 4b). In line with this, repression of ROR1 in SNU387 cells with the mesenchymal phenotype caused even more expression of the canonical EMT marker vimentin (Figure 4c). SNU387 cells do not express any E-cadherin therefore we did not investigate its expression [33]. These results suggest that the decreased expression of ROR1 in HCC cells during TGF-β induced EMT or its targeted silencing is not a bystander of the trans-differentiation program, but functionally contributing to EMT phenotype.

### 3.5. ROR1-Depleted HCC Cells Display Reduced Motility and Proliferation and Increased Accumulation at G1 Phase

Developmental EMT proceed with several important hallmarks. Firstly, cells lose their epithelial characteristics such as decreased keratin, E-cadherin and other epithelial markers. Secondly, they acquire mesenchymal features such as increased expression of N-cadherin and vimentin. This translates into enhanced migration/invasion capability of cells undergoing EMT [36,37]. Additionally, EMT was also shown to induce cell cycle arrest, resistance to apoptosis, and stem cell features [38]. To investigate the contribution of ROR1 towards the above listed hallmarks of EMT, we first assessed the migration capabilities of ROR1-knockdown and control clones of PLC and SNU387 cells. In a transwell migration assay, we detected decreased migration of cells upon ROR1 depletion (Figure 5a). This unexpected result prompted us to inquire whether cellular proliferation and cell cycle progression were altered depending on ROR1 expression. Compared to control, ROR-knockdown clones of SNU387 and PLC cells showed significantly reduced proliferation (Figure 5b). This was in line with the accumulation of ROR1-depleted cells at G1 phase (Figure 5c). Therefore, decreased ROR1 expression positively contributes to the cell cycle arrest phenotype of EMT but unexpectedly reduces motility.

### 3.6. Decreased ROR1 Confers Resistance to Apoptosis and Anoikis on HCC Cells

Another key hallmark of EMT is resistance to apoptosis [39]. Systemic Doxorubicin administration or trans-arterial embolization coupled platinum derivatives are commonly used in the treatment of HCC [40,41]. To investigate if ROR1 contributes to chemotherapy response, and therefore to apoptosis, we treated PLC-pLKO, PLC-shROR1, SNU387-pLKO, and SNU387-shROR1 cells with increasing concentrations of Doxorubicin and Oxaliplatin. Analysis of survival through uncleaved PARP (p116) showed that PLC cells with decreased ROR1 (PLC-shROR1) died significantly less (*p* < 0.05) upon pro-apoptotic stimuli (Figure 6a). However, neither SNU387-pLKO nor SNU387-shROR1 displayed sensitivity to Oxaliplatin doses to which PLC control and ROR1-knockdown cells were responsive (Appendix A
Figure A2A). But, a similar trend of chemoresistance, albeit almost reaching significance (*p* = 0.12), to Doxorubicin was observed in SNU387-shROR1 cells (Appendix A
Figure A2B). Of note, mesenchymal HCC cells such as SNU387 are more chemoresistant than epithelial ones (such as PLC cells) therefore it is only conceivable to observe that chemotherapeutic drug concentrations to kill these cell types should be vastly different [42].

Another important aspect of EMT is anchorage-independent growth-induced cell death (anoikis). It was significantly reduced (*p* < 0.05) in both ROR1-depleted PLC and SNU387 cells as compared to their respective controls (Figure 6c and Appendix A
Figure A2C). Anoikis resistance is a key aspect of cancer stem cell behavior and commonly assessed during cancer stem cell assays [43]. These results suggest that, unlike other cancers, decreased ROR1 is necessary for acquisition of certain aspects of mesenchymal phenotype in HCC including resistance to cell death induced by DNA damaging agents and contact inhibition.

### 3.7. ROR1 Depletion Induces Resistance to Chemotherapeutic Drugs through Upregulation of Multidrug Resistance Genes

Mesenchymal/metastatic carcinoma cells are known to be resistant to different forms of cell death including apoptosis and anoikis [43]. The mechanisms of this occurrence are relatively unknown; however, increased drug efflux, improved DNA repair, rewiring of pro survival signaling, attenuated DNA damage response, and pro-apoptotic signaling have thus far been suggested as contributing factors [39,44,45,46]. As ROR1 depletion induced an apparent resistance to chemotherapeutic agent-induced apoptosis in PLC cells, as shown using selected concentrations of Doxorubicin and Oxaliplatin (Figure 6a,b), we investigated whether drug uptake is regulated by ROR1. To achieve this aim, the fluorescence properties of Doxorubicin was used. PLC-pLKO and PLC-shROR1 cells were incubated with Doxorubicin for 1 h, washed and analyzed for the presence of retained Doxorubicin 4 h later using flow cytometry. Both cells showed increased fluorescence signal at the correct channel compared to untreated cells. Importantly, PLC-shROR1 cells registered significantly less (*p* < 0.05) intracellularly retained Doxorubicin suggesting enhanced drug efflux upon ROR1 depletion (Figure 7a). To explore the mechanism of ROR1-knockdown-mediated drug efflux, we probed for the expression of critical multidrug transporter genes. Our results showed significant upregulation in *ABCB1*, which is also known as multidrug resistance 1 (*MDR1*), *ABCC6,* and *ABCG1*, which were associated with cancer chemoresistance previously [47,48,49,50] (Figure 7b). Our findings related to increased expression of multidrug resistance genes upon ROR1 knockdown provide a mechanistic explanation towards the chemoresistance phenotype observed.

## 4. Discussion

HCC remains one of the very few cancers where incidence remains high, and despite the recent advances in therapeutic options, the prognosis of patients is still poor [51,52,53]. According to Cancer Research UK predictions, the mortality rate of HCC will increase up to 43% within the next 20 years [54]. The risk factors implicated in HCC are very well defined so, in theory, potential patients who will develop the disease should be easy to identify. However, almost all cases of HCC develop on chronic liver insult leading to fibrosis and cirrhosis, which complicates diagnostic guides [55]. For those reasons, identifying and discriminating early HCC from underlying liver pathologies is important. Previous studies showed that ROR1 is not expressed in normal liver or in normal hematopoietic cells in blood [24,56]. In this study, we identified easily detectable ROR1 protein expression in a panel of human and mouse HCC cell lines. Therefore, given the absence of ROR1 protein in normal blood cells, identifying ROR1 protein expressing cells in blood, e.g., in the context of circulating tumor cells or free protein, would serve as a good diagnostic indicator of HCC, provided ROR1 expression is not upregulated in different kinds of non-malignant liver pathologies [16].

In this study, we showed that ROR1 protein is overexpressed in hepatocellular carcinoma-derived cells from both mice and humans. In this regard, HCC is similar to pancreatic cancer where almost all cell lines expressed ROR1 but with minor differences in abundance [57]. On the other hand, in certain cancers such as breast, ovary, and lung, ROR1 protein is restricted to cells showing mesenchymal/metastatic features [20,31,58]. As a matter of fact, ROR1 was identified as one of the few kinases specifically upregulated in a well-defined model of breast cancer stem cell phenotype [59]. Surprisingly, we found an EMT like phenotype in HCC cells upon shRNA-mediated down-regulation of ROR1. The epithelial HCC cell line, PLC/PRF/5, showed increased vimentin and decreased E-cadherin, which are considered hallmarks of EMT under ROR1-depleted conditions. In addition, these cells displayed decreased the expression of another epithelial marker, cytokeratin 19. Moreover, the mesenchymal HCC cell line SNU387 displayed augmented vimentin expression upon targeted repression of ROR1. We also investigated other hallmarks of EMT such as increased motility, cell cycle arrest, and resistance to anoikis and apoptosis. Downregulation of ROR1 went hand in hand with features of EMT, including cell cycle arrest and resistance to cell death but not with increased motility in HCC cells. To our best knowledge, this is the first description that cells displaying most of the EMT characteristics lost or have not yet gained their ability of enhanced migration. Further, our results are in line with recent findings that the EMT is not a binary switch between fully epithelial and fully mesenchymal states but rather involves multiple stages that can be expressed between these two culminated phenotypes of the EMT program [60]. For instance, in a non-carcinoma context, ROR1 was shown to be decreased during acquisition of metastatic features. During progression from dysplastic naevi to malignant melanoma, ROR1 expression goes down and the rate of cell division decreases in accordance with metastatic phenotype [61]. Therefore, and considering the contribution of ROR1 to EMT in melanoma and as shown here in HCC, it is plausible to hypothesize that ROR1 activity can be a decisive point during carcinogenesis. In our case, ROR1-knockdown induced chemoresistance in HCC cells against conventional chemotherapeutic agents used in HCC treatment, such as systemically administered Doxorubicin and the platinum derivative, Oxaliplatin, which is used in trans-arterial chemoembolization (TACE) [40,41]. In that regard, activation rather than inhibition of ROR1 may be important to retain the chemosensitive nature of HCC cells. Therefore, testing HCC tumors for their ROR1 expression might be critical in the clinical practice to decide whether chemotherapy would be beneficial for the patients or not.

Our results suggest that chemoresistance acquired by the decrease of ROR1 is mediated, at least partially, via expulsion of drugs, which in turn is due to upregulation of multidrug resistance genes. In this context, it is important to mention that the signaling pathways acting upon ROR1 activation or repression have not been clearly identified. Our results will aid in clarification of cellular pathways working under ROR1, as well as defining the signaling network in HCC, especially when ROR1 is downregulated. The emergence of MDR genes as downstream effectors of ROR1 signaling will potentially lead to the further characterization of the ROR1 pathway. In conclusion, the fact that ROR1 downregulation leads to EMT phenotype adds a new layer of complexity to the understanding of metastasis of HCC.

## Figures and Tables

**Figure 1 cells-08-00210-f001:**
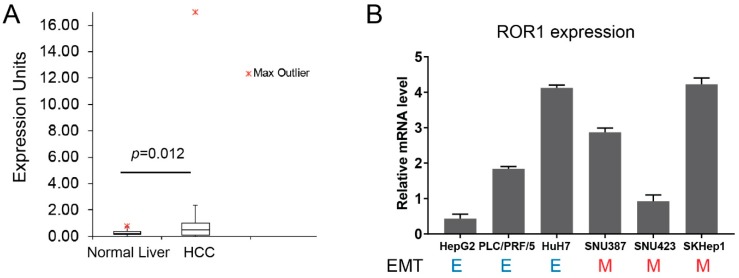
Detection of ROR1 expression in hepatocellular carcinoma (HCC) cells. (**A**) Gene expression of *ROR1* in vivo. Box plot shows the expression of *ROR1* in human HCC samples (*n* = 99) as compared to normal liver tissues (*n* = 52). Tumor tissues express significantly more ROR1 (*p* = 0.012). (**B**) Expression of *ROR1* in mRNA level was detected in HCC cells by RT-qPCR. *ROR1* expression was calculated via normalization with *GAPDH*, and represented as a relative mRNA level. E: epithelial and M: mesenchymal.

**Figure 2 cells-08-00210-f002:**
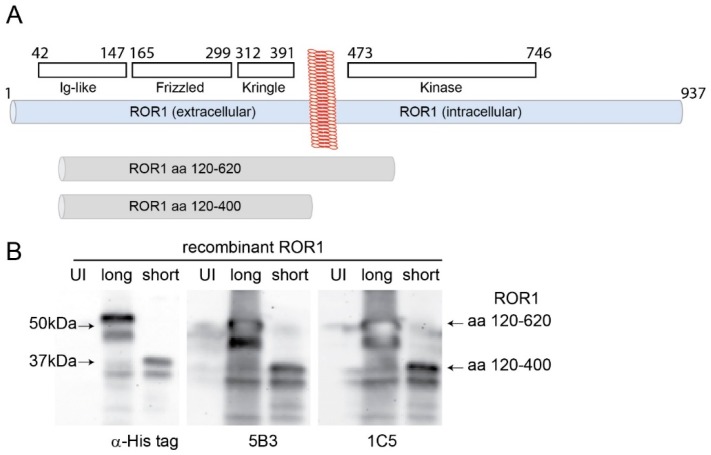
Generation monoclonal antibodies against ROR1. (**A**) Schematic representation of the domains of ROR1. Recombinant protein of ROR1 (aa 120–620) was used for the immunization. Recombinant protein of ROR1 (aa 120–400) was generated with site-directed mutagenesis and represents the extracellular part of ROR1 protein. (**B**) Western blot analysis of ROR1 antibodies (5B3 and 1C5) and His-tag antibody. Both 5B3 and 1C5 antibodies detected the both full and short length recombinant proteins of ROR1.

**Figure 3 cells-08-00210-f003:**
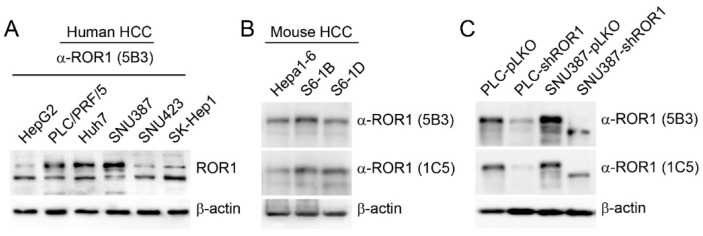
Expression of ROR1 in human and mouse HCC cell lines. (**A**) Western blotting analysis of ROR1 expression in human HCC cells. ROR1 was detected with 5B3 monoclonal antibody. (**B**) Western blotting analysis of ROR1 expression in mouse HCC cells. Both 5B3 and 1C5 antibodies were used to detect ROR1 in mouse HCC cell lines. (**C**) Specificity of ROR1 antibodies was tested in shRNA ROR1-knockdown and control clones. β-actin was used as a loading control.

**Figure 4 cells-08-00210-f004:**
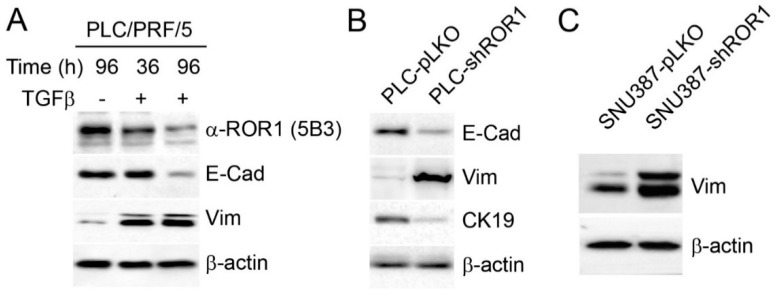
Knockdown of ROR1 is associated with the mesenchymal phenotype. (**A**) PLC/PRF/5 cells were treated with 4 ng/mL TGF- β for 36 and 96 h. Western blotting shows the expression of ROR1, E-cadherin and vimentin. (**B**) Analysis of the expression of mesenchymal phenotype related proteins in PLC-shRNA ROR1-knockdown and control clones. (**C**) Expression of vimentin in 387-shRNA ROR1-knockdown and control clones. β-actin was used as a loading control.

**Figure 5 cells-08-00210-f005:**
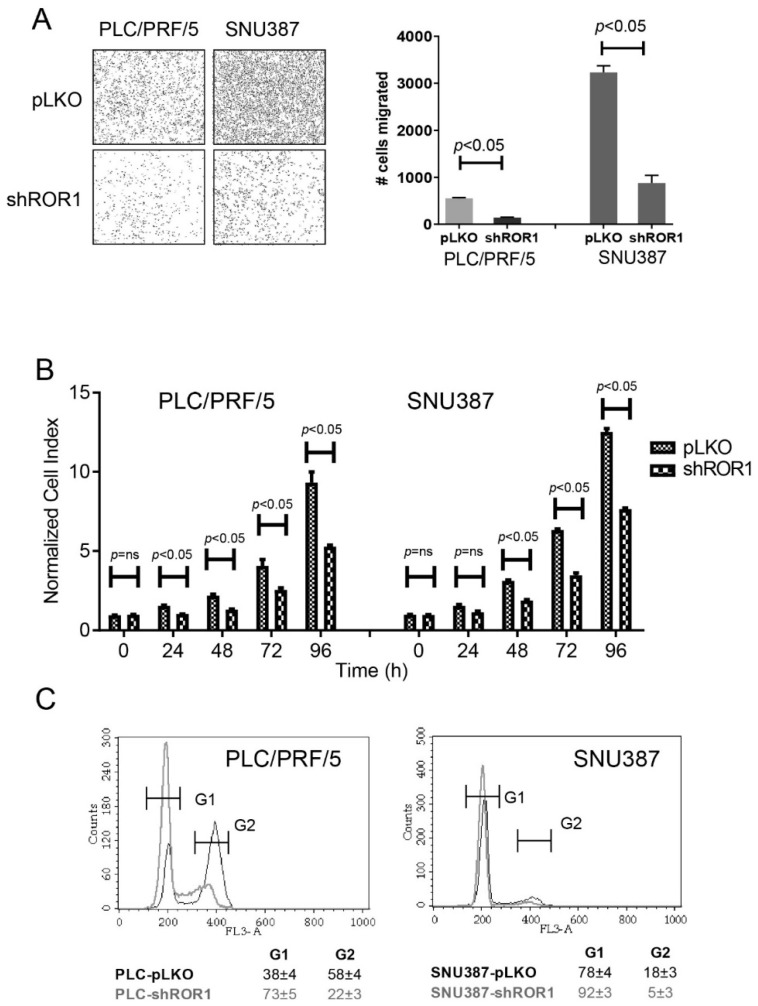
Analysis of cell cycle, proliferation, and migration capacities of shRNA ROR1-knockdown and control clones. (**A**) ROR1-knockdown and control clones were seeded into insert chambers of transwell plates and were allowed to migrate for 48 h. The number of migrated cells were counted by using ImageJ analysis of stained insert membranes. (**B**) Proliferation of ROR1-shRNA knockdown and control clones was analyzed by xCELLigence real-time cell analysis (RTCA) systems. Proliferation of cells was represented as a normalized cell index. *** *p* < 0.001. (**C**) Propidium iodide DNA staining was used to analyze the cell cycle state of ROR1-knockdown and control clones of SNU387 and PLC cells by flow cytometry.

**Figure 6 cells-08-00210-f006:**
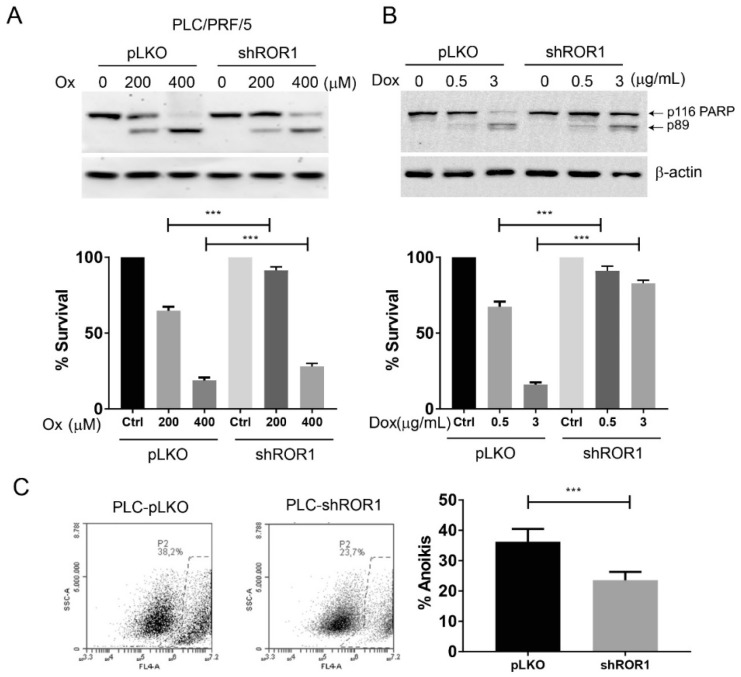
Analysis of resistance to apoptosis and anoikis in shRNA ROR1-knockdown and control clones. (**A**) PLC-shRNA ROR1-knockdown and control clones were treated with various concentration of Oxaliplatin for 24 h. PARP antibody was used to detect apoptosis. Protein band intensity of uncleaved PARP was measured by ImageJ to detect the ratio of surviving cells under drug treatment. (**B**) PLC-shRNA ROR1-knockdown and control clones were treated with various concentration of Doxorubicin for 24 h. PARP antibody was used to detect apoptosis. Protein band intensity of uncleaved PARP was measured by ImageJ to detect the ratio of surviving cells under drug treatment. β-actin antibody was used as a loading control. (**C**) shRNA ROR1-knockdown and control clones were seeded into ultra-low attachment plates and the ratio of apoptotic cells was calculated after 24 h via flow cytometry by using annexin V antibody. *** *p* < 0.05.

**Figure 7 cells-08-00210-f007:**
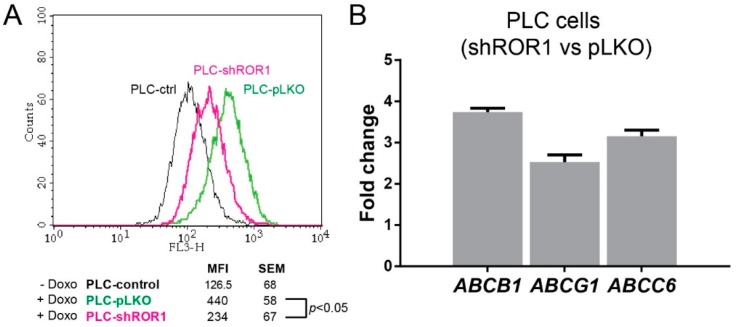
Analysis of drug resistance of shRNA ROR1-knockdown and control clones. (**A**) Drug efflux capacities of shRNA ROR1-knockdown and control clones: Cells were treated with 0.5 μg/mL Doxorubicin for 1 h and mean fluorescence intensities (MFI) were measured with flow cytometry after 4 h. SD: standard deviation. (**B**) Expression of ABC genes in mRNA levels were analyzed in shRNA ROR1-knockdown clones and calibrated to control clones by RT-qPCR. Relative fold change values (log2) were normalized to *GAPDH*.

**Table 1 cells-08-00210-t001:** List of primers for PCR amplification of drug resistance genes.

Gene	F (5′–3′)	R (5′–3′)
ABCG1	CTACCACAACCCAGCAGATT	AGGTCTCTCTTGTGGTCTGA
ABCC6	CCCAGGCTGATTGGATCATAG	GGTTCTGTTTCTCCTTCTCCTC
ABCB1	GGTGAGTCAGGAACCTGTATTG	AGTCATAGGCATTGGCTTCC

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
