# Peer review of "ROR1 Expression and Its Functional Significance in Hepatocellular Carcinoma Cells"

_cells, 2019, doi:10.3390/cells8030210_

Round 1
Reviewer 1 Report
I think that this new version of the manuscript is quite better than the previous one and responds to most of my concerns. Thus, I consider it acceptable for publication.
Reviewer 2 Report
I appreciate the efforts the authors made to respond to the reviewer's comments.
This manuscript is a resubmission of an earlier submission. The following is a list of the peer review reports and author responses from that submission.
Round 1
Reviewer 1 Report
In this study, the authors investigated the expression of receptor tyrosine kinase-like orphan receptor 1 (ROR1) in hepatocellular carcinoma (HCC) cell lines and its functional role in biology of these cell lines. For this purpose, they generated two clones of monoclonal antibodies for ROR1 and validated its immunoreactivity. They demonstrated that mRNA and protein for ROR1 were detectable in all of the tested HCC cell lines. Downregulation of ROR1 with shROR1 was associated with changes of gene expression concordant with epithelial-to-mesenchymal transition (EMT). However, knockdown of ROR1 unexpectedly decreased proliferation and migration capacities, which was discordant with the feature of EMT.
The manuscript is clearly written but findings as well as significance of this study are not satisfactory.
The title of the manuscript stresses the generation of new antibodies for ROR1. However, there are several ROR1 antibody products that are commercially available. Also, they just tested immunoreactivity of the antibodies in Western blotting using HCC cell lines. If the authors like to emphasize the generation of the new antibodies, they should test them in the other uses such as immunostainings and immunoprecipitation not only in cell lines but also in tissues. Agonistic or antagonistic activities should be tested as well.
Actually I would suggest the authors to change the title so that it does not stress the generation of the new antibodies.
I would suggest analysis of downstream signaling pathways of ROR1 such as hematopoietic-lineage-specific protein 1 (HS1) and RhoA.
How can the authors explain reasons for the discrepancy between EMT-related gene expression (downregulation of E-cadherin and upregulation of vimentin) and decreased proliferation and migration capacities upon knockdown of ROR1 ?
What does the overexpression of ROR1 result in ?
In introduction and discussion, the authors emphasized that there are scarce therapeutic options for HCC. The authors would be cynical about this description because nowadays many agents have been available for HCC such as sorafenib, regorafenib, and lenvatinib and the trials of many other drugs are underway such as nivolumab and pembrolizumab.
Minor points
page 5, line 16
5 min minutes -> 5 minutes
Reviewer 2 Report
In this paper, Cetin et al generate new specific anti-ROR1 antibodies. They further show that ROR1 knockdown in one hepatocarcinoma cell line resulted in decreased proliferation and migration, but enhanced resistance to apoptosis and anoikis. This chemotherapy-resistant phenotype of ROR1 knockdown cells seems to be due to enhanced drug efflux and it is associated with modulation of expression of several multi-drug resistance genes.
Major concerns,
1. The experiments are well performed and the data obtained is of interest, but this study has important limitations. The most important one is the fact that most of the data have been obtained from only one hepatocarcinoma cell line and there are no data of the expression of ROR1 in normal tissues or in patients´samples. Thus, it is no clear whether an aberrant expression of ROR1 in hepatocarcinoma really exists, as the authors stated; and it is not clear whether the results obtained are only a specific characteristic of this cell line or they really reflect the behavior of this kind of tumor. The repetition of the experiments with more cell lines and the analysis of the expression of ROR1 in patient´s samples are essential to establish the role of ROR1 in this type of tumor.
2. The authors have generated anti-ROR1 antibody, which is of interest, but there is a poor characterization of their characteristics. For instance, flow cytometry analyses could be easily performed to characterize if these antibodies may recognize the native form of ROR1, which is clearly essential to establish the relevance of these antibodies.
3. The authors stated that the ROR1 knockdown phenotype was associated with enhanced drug efflux and with the induction of expression of several multi-drug resistance genes, providing an underlying mechanisms for the resistance of these cells to apoptosis. In agreement, an increase of the expression of some multi-drug resistance genes is observed in figure 7, but there is also a reduction of the expression of many others being no clear which could be the final effect of these genes in the drug efflux.
4. In base of the results provided, the role of ROR1 in the EMT process in hepatocarcinoma is very questionable.
Reviewer 3 Report
Manuscript # 439250
Title: Assessment of New ROR1 Antibodies for Investigating Metastatic Attributes of Hepatocellular Carcinoma Cells.
Corresponding Authors: Emre Sayan and Tamer Yagci
Contributing Authors: Metin Cetin, Gorkem Odabas, Leon R. Douglas, Patrick J. Duriez, Pelin Balcik-Ercin , Irem Yalim-Camci
This manuscript has been submitted by the group of experts in metastasis of solid tumors.
To date, there is little progress in the treatment of Hepatocellular Carcinoma Cells (HCC). In this manuscript, the authors investigate the ROR1 role in EMT and metastatic attributes of HCC.
The authors have analysed a set of HCC cell lines characterised by different EMT status and found differential expression of ROR1. This type I membrane protein typically found in early embryonic development but expression mostly lost in adult tissues. The authors found that on mRNA level the differential expression of ROR1 did not correlate with different EMT phenotypes of the HCC cell lines. Both the epithelial and mesenchymal types of HCC cell lines have shown differential expression of ROR1. Next, the authors set out to create a new bunch of anti-ROR1 antibodies using a unique strategy of epitope selection, and immunisation eventually selecting two clones of antibodies with strong ROR1 binding and cross-species reactivity for human and mouse ROR1. Next, the authors confirm the specificity of the new antibodies comparing ROR1 protein expression in WT cell lines and ROR1 knockdown with ROR1-shRNA. After that, the authors address the question of how the ROR1 expression influence the metastatic phenotype of the HCC cell lines using TGFb activation of EMT and ROR1 knockout. They found that the activation of EMT goes side by side with a decrease of ROR1 protein expression and an increase of vimentin and reduction of E-cadherin expression. Knockdown of ROR1 without any additional stimulation also causes similar changes, increase in vimentin and a decrease in E-cadherin expression. After that, the authors checked how ROR1 knockdown affect HCC cell line motility, proliferation and cell cycle. They found that a reduction of ROR1 protein expression causes the decrease in cell motility in transwell experiment, a decrease in proliferation and accumulation of cells in G1 phase. Next, the authors addressed the question on the role of ROR1 in resistance to apoptosis and anoikis and found that a decrease in ROR1 expression increases cell survival. Following experiments have shown that ROR1 depletion increases the expression of genes involved in drug resistance notably ABCB1 (MDR1) and ABCG1.
The role of the aberrant expression of ROR1 by tumor cells attracts a lot of interest lately, especially recent observation of ROR1 aberrant expression in several solid tumors and some blood malignancies. Indeed, in Chronic Lymphocytic Leukaemia where 95% of cases are ROR1 positive, anti-ROR1 monoclonal antibodies are already in clinical trials (Cirmtuzumab). The main novelty of this manuscript is the observation of aberrant expression of ROR1 in HCC and its role in the regulation of the metastatic phenotype. Interestingly, the authors raised a set of new anti-ROR1 monoclonal antibodies using crafty the antigen epitope selection and immunising strategy. The results of this study are important and pave the way for prognosis and development of future HCC treatment. The authors need to address some minor problems with the text to ensure the acceptance of this manuscript.
General remark: the authors need a careful proofreading, some of the sentences need to be rewritten in a more reader-friendly way.
Here some recommendations:
Page 2 line 31. “Alterations of various pro-survival..” please specify what alterations? gene mutations, epigenomic or signalling etc.
Page 3 line 29. “ROR1, as an RTK is not expressed in adult tissues including brain …” Better to state about its expression in embryogenesis or only limited expression in some adult tissues.
Page 3 line 31. “and its aberrant expression in HCC has not been studied so far. For that reason, creation of new antibodies and assessment of ROR1 involvement in this particular cancer is of utmost importance.” This is an example of the weak reasoning, please rewrite and make your argument stronger, e.g. the HCC metastasises are EMT driven, in other solid tumors the role of ROR1 in EMT has been shown but no one studied this in HCC etc. Also unclear why would you need a set of new antibodies? e.g. Cirmtuzumab is already in clinical trials? Other Abs commercially available. Is it due to your specific strategy of raising a monoclonal antibody that can deliver better specificity/affinity of your antibodies? Please explain in one sentence.
A general remark about Introduction. There are anti-ROR1 antibodies in clinical trials at the moment, and you discuss this in the Discussion section but there is no mention of it in the introduction (e.g Kipps or Choi papers).
Page 4 line 2. “ROR1 knockdown altered EMT status” please specify.
Page 4 line 10. “SP2/0 cells” - cell lines
Page 6 line 9-10. “uptake, Migration, Drug resistance” - uptake, Migration and Drug resistance.
Page 7 line 5-9. At this point it is unclear what have you measured, only later in the text you explained that Doxorubicin is fluorescent, please mention this in the materials.
Page 7 line 20. What is “macro”? If you mention, please give a full name and company name.
Page 8 line 11-13. These lines are written in the way which contradicts your hypothesis. Obviously, your initial study was on an mRNA level, and you did not see mRNA ROR1 correlation with EMT features which lead you to the next stage of this study looking for protein expression.
Page 8. Figure 1. It is unclear what was used to set a baseline, if you calculate a relative expression it has to be related to something. e.g. to the cell line with lowest mRNA expression.
Page 9 line 9. “we performed hydrophobicity analysis” but did not explain why your epitope is unique, what features make it so and why it is essential for the generation of antibody. General remark: the generation of novel antibodies is stated as one of the main novelty of this manuscript (title) so one would expect to see more explanation on the strategy of epitope selection, immunisation, resulting antibodies specificity and affinity and advantages of your approach over others.
Page 10 line 3. ROR – ROR1
Page 10 line 15. “These results suggest” - These results confirm
Page 10 Figure 3. Please mention how the mRNA results correlate with protein expression in these six cell lines.
Page 11 Figure 4. It is unclear why the authors have chosen PLC/PRF/5 cell line which is not strongest ROR1producer. Please explain why you did not use Huh7?
Page 14 Figure 6 and Page 15. It is unclear how the dosage and time of treatment were established. Previous publications?
Page 16 line 21. “Therefore, identifying ROR1 protein expressing cells in blood would serve as” any ROR1 positive cells? Is it circulating HCC cells? Did anyone look for ROR1 positive circulating cells in a healthy individual?
Discussion. Again, the authors stated that the creation of novel anti-ROR1 antibodies was one of their achievement, one would expect this would be mentioned in the discussion e.g. comparison to existing antibodies.
Page 17 line 32-36. Please re-write this sentence.
Page 18 line 1-3. Explain why it has not been done in this manuscript.